# Application of attenuated total reflection–Fourier transform infrared spectroscopy in semi-quantification of blood lipids and characterization of the metabolic syndrome

Tz-Ping Gau[1,2,3], Jen-Hung Wen[4], I-Wei Lu[5], Pei-Yu Huang[6], Yao-Chang Lee[6,7,8], Wei-Po Lee[3], Hsiang-Chun Lee[9,10,11] *

1 Department of Anesthesiology, Kaohsiung Medical University Hospital, Kaohsiung, Taiwan, 2 Center for Big Data Research, Kaohsiung Medical University, Kaohsiung, Taiwan, 3 Department of Information Management, National Sun Yat-sen University, Kaohsiung, Taiwan, 4 School of Medicine, College of Medicine, Kaohsiung Medical University, Kaohsiung, Taiwan, 5 Center for General Education, Kaohsiung Medical University, Kaohsiung, Taiwan, 6 Group of Life Science, National Synchrotron Radiation Research Center, Hsinchu, Taiwan, 7 Department of Optics and Photonics, National Central University, Chung-Li, Taiwan, 8 Department of Chemistry, National Tsing Hua University, Hsinchu, Taiwan, 9 Division of Cardiology, Department of Internal Medicine, Kaohsiung Medical University Gangshan Hospital, Kaohsiung, Taiwan, 10 Lipid Science and Aging Research Center, Kaohsiung Medical University, Kaohsiung, Taiwan, 11 Department of Internal Medicine, School of Medicine, College of Medicine, Kaohsiung Medical University, Kaohsiung, Taiwan

* hclee@kmu.edu.tw

## Abstract

### Background/Purpose

Dyslipidemia, a hallmark of metabolic syndrome (MetS), contributes to atherosclerotic and cardiometabolic disorders. Due to days-long analysis, current clinical procedures for cardiotoxic blood lipid monitoring are unmet. This study used AI-assisted attenuated total reflectance Fourier transform infrared (ATR-FTIR) spectroscopy to identify MetS and precisely quantify multiple blood lipid levels with a blood sample of 0.5 μl and the assaying time is approximately 10 minutes.

### Methods

ATR-FTIR spectroscopy with 1738 data points in the spectral range of 4000–650 cm$^{-1}$ was used to analyze the blood samples. An adaptive synthetic technique was used to establish a prevalence-balanced dataset. LDL-C, HDL-C, TG, VLDL-C, and cholesterol levels were defined as the predicted targets of lipid absorption profiles. Linear regression (LR), gradient boosting regression tree (GBT), and histogram-based gradient boosting regression tree (HGBTR) were used to train the models. Lipid profile value prediction was evaluated using $R^2$ and MAE, whereas MetS prediction was evaluated using area under the ROC curve.

**Data availability statement:** All relevant data are within the paper and its Supporting information files.

**Funding:** We received funding to support this study and updated the Funding Information on the Submission page as follows: National Health Research Institutes (NHRI-EX107-10724SC, NHRI-EX108-10724SC, NHRI-EX109-10724SC, and NHRI-EX110-10724SC), Kaohsiung Medical University Hospital (KMUH111-1R07 and KMUH110-0R11), Taiwan Ministry of Science and Technology/ National Sciences, and Technology Council Grants (MOST 109-2314-B-037-111-MY3) NSTC 112-2314-B037-069. The funders had no role in study design, data collection and analysis, decision to publish, or preparation of the manuscript.

**Competing interests:** The authors have declared that no competing interests exist.

## Results

A total of 150 blood samples from 25 individuals without MetS and 25 with MetS yielded 491 spectral measurements. In the regression models, HGBT best predicted the targets of TG, CHOL, HDL-C, LDL-C, and VLDL-C with $R^2$ values of 0.854 (0.12), 0.684 (0.08), 0.758 (0.10), and 0.419 (0.11), respectively. The classification model with the greatest AUC was RF (0.978), followed by HGBT (0.972) and GBT (0.967).

## Conclusion

The results of this study revealed that predicting MetS and determining blood lipid levels with high $R^2$ values and limited errors are feasible for monitoring during therapy and intervention.

## Introduction

Metabolic syndrome (MetS) is the leading cause of cardiometabolic diseases and has become a significant health concern worldwide [1,2]. According to the National Cholesterol Education Program (NCEP), MetS is diagnosed based on elevated glucose levels, reduced HDL cholesterol, elevated triglycerides (TG), obesity, and hypertension [3]. Beyond low-density lipoprotein cholesterol (LDL-C), TG-rich lipoproteins, such as very-low-density lipoprotein (VLDL), have been shown to cause cardiac remodeling [4–6], particularly by inducing atrial dilation, which is a hallmark of atrial myopathy and represents a risk for atrial fibrillation (AF). Other hazardous lipids, such as diacylglycerols and ceramides, can promote atrial myopathy by promoting mitochondrial dysfunction and apoptosis [7]. Some saturated free fatty acids can affect cardiac ion channels by activating macrophages and IL-6 [8], and postprandial VLDL is correlated with atrial remodelling [5], suggesting that dietary fat and postprandial circulatory lipids can affect arrhythmogenicity [9]. Moreover, VLDL is toxic to the cardiovascular system and can induce vulnerability to AF in patients with MetS but not in healthy subjects [10,11].

Circulatory lipids, except for some albumin-binding fatty acids, are carried by lipoproteins. VLDL lipotoxicity is primarily affected by the lipid core, which consists of a variable abundance of lipid species. Quantification of lipid species in lipoproteins can be performed using nuclear magnetic resonance (NMR), mass spectrometry (MS), gas chromatography (GC), and Fourier transform-ion cyclotron resonance (FT-ICR) [12]. However, these costly and time-consuming methods are not feasible for daily applications in clinical scenarios. Moreover, none of them can be used as a monitoring assay when repeated testing is required, for instance, when assessing blood lipid changes after a nutritional intervention or exercise workout.

Attenuated total reflectance Fourier transform infrared (ATR-FTIR) spectroscopy is a technique for analyzing the chemical composition of materials by detecting their characteristic absorption of mid-infrared light. In ATR-FTIR, molecules absorb mid-infrared light at specific wavenumbers (or frequencies), depending on their chemical bonds and structure. Each type of bond (e.g., C=O and C–H) has a unique absorption pattern, creating a molecular fingerprint. ATR-FTIR can quickly analyze the molecular fingerprint of an organic sample and reveal its principal biomolecular composition and interactions and has been validated in a broad spectrum of biomedical applications, such as SARS-CoV-2 detection [13,14], urinalysis for assessing renal hyperfiltration [15], and the detection of HIV infection [16]. The advantages of FTIR, with its quick assaying time and demand for tiny blood samples (0.5 μL,

equivalent to a finger-prick sampling amount) prove that it is an ideal method for blood lipid analysis.

In ATR-FTIR, lipids in the blood samples produce clustered signals representing IR reflections from structures with a range of molecules conjugated to various proteins. Conventional algorithms for spectral analysis cannot categorize lipid species. Algorithms are used to directly estimate and quantify the amount of specific lipids to generate raw reflection data with complex interactions. Therefore, an effective analytical method for ATR-FTIR is required to facilitate its application in lipid analyses.

Data-driven processing enables rapid development and application of artificial intelligence (AI). The introduction of the gradient boosting tree model (GBT), linear regression model (LR), and histogram-based gradient boosting tree model (HGBT), an enhanced variant of the GBT model, makes machine learning useful in complex biomedical prediction. Some examples include the prediction of progression-free survival in prostate cancer [17], estimation of delivery date in pregnancy [18], and evaluation of one-year mortality in patients receiving coronary stenting [19]. AI-assisted ATR-FTIR spectroscopy has great potential as a rapid, economical, and resilient diagnostic platform for detecting and monitoring metabolic disorders. Thus, it can be widely used in the clinical setting. In this study, we combined machine learning and ATR-FTIR data to develop a valuable method for distinguishing between patients with and without MetS and for predicting conventional laboratory test results.

## Methods

### Setting and participants

The study protocol was approved by the Kaohsiung Medical University Hospital Institutional Review Board (KMUHIRB-F(II)-20210091) and was implemented in accordance with the Good Clinical Practice Guidelines, the principles of the Declaration of Helsinki, and all applicable legislation and regulations of the Kaohsiung Medical University Hospital. This retrospective study accessed data from 2021/05/25, and all authors had access to information that could identify individual information after data collection. All study participants provided their informed written consent. Two blood samples were collected from each subject and centrifuged to keep the serum frozen for the ATR-FTIR experiments. Clinical and laboratory data were obtained. The study participants were divided into two groups based on the presence or absence of MetS. The clinical diagnosis of MetS is defined by the existence of at least three criteria as follows: waist circumference (greater than or equal to 90 cm in men and greater than or equal to 80 cm in women), blood pressure ($\geq$ 130/85 mmHg), blood glucose level (fasting state greater than or equal to 100 mg/dl), TG (greater than 150 mg/dL), and high-density lipoprotein cholesterol (HDL-C level, for male less than or equal to 40 mg/dL, for female less than or equal to 50 mg/dL).

### ATR-FTIR spectral data acquisition

All ATR-FTIR spectral data were obtained at Taiwan Light Source (TLS) BL14A1 of the National Synchrotron Radiation Research Center (NSRRC) in Taiwan. This study applied a small sample volume (0.5 μL to a Ge ATR crystal, as shown in Fig 1(A). A modulated mid-infrared beam was directed into the Ge crystal, generating an evanescent wave at the crystal surface, allowing the light to interact with the sample. The sample absorbs light at characteristic wavenumbers, and the remaining light is detected using a liquid nitrogen-cooled mercury cadmium telluride (MCT) infrared detector. The resulting spectrum shows peaks corresponding to specific molecular bonds, providing a unique chemical profile of the sample. ATR-FTIR samples must be stored in a −80 °C freezer to ensure sample stability before measurement

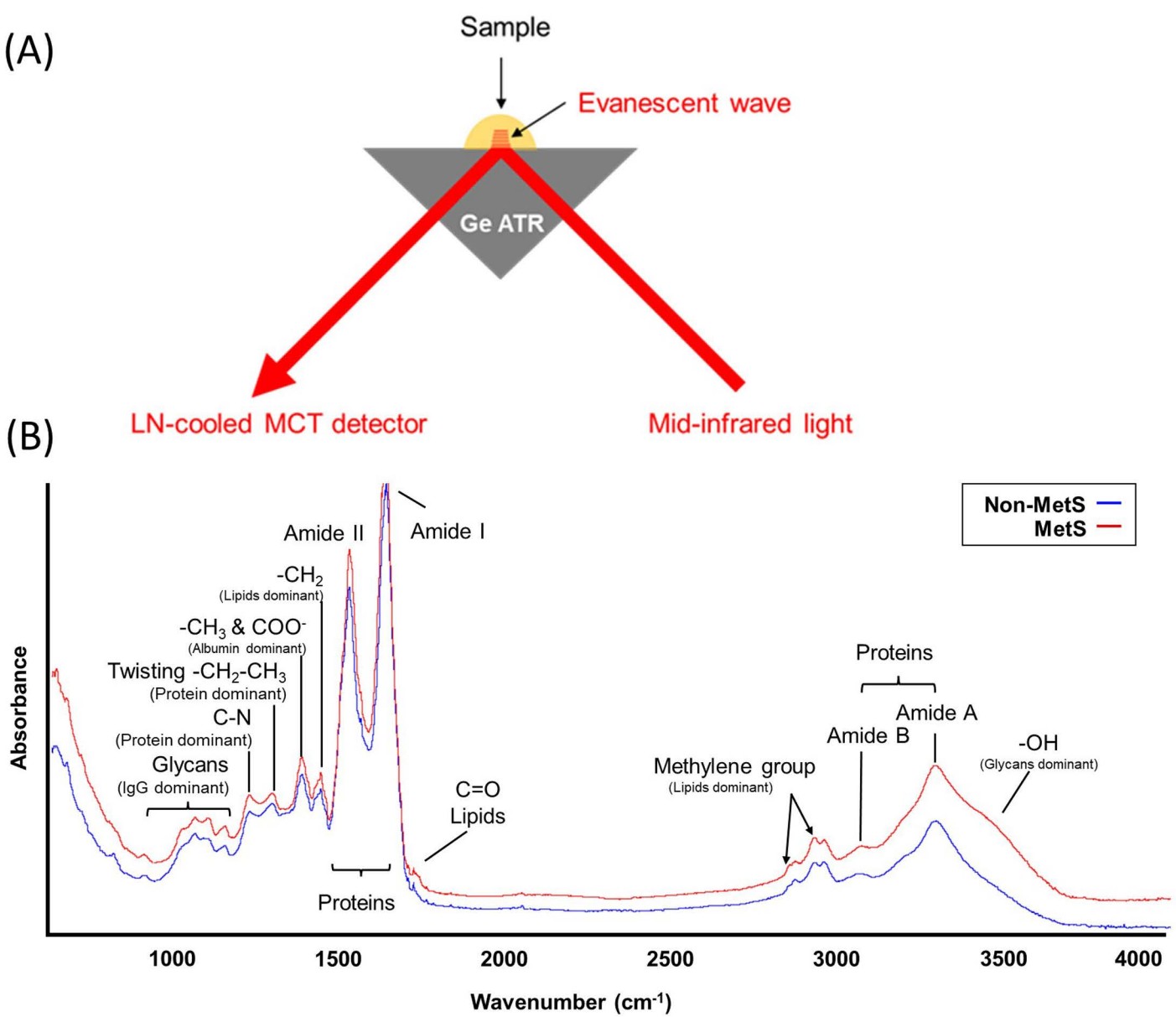

**Fig 1.** (A). The schematic depicts the attenuated total reflection for acquiring the FTIR spectra of a blood sample. The mid-infrared light is modulated using the Michaelson interferometer of an FTIR spectrometer. (B). **Representative ATR-FTIR spectrometry.** Raw data are obtained from plasma samples from one MetS and one non-MetS individual. The absorption regions of the major biomolecules are indicated.

and brought to room temperature naturally before measurement. The samples were placed on the ATR crystal surface to allow moisture to evaporate naturally before the signal acquisition, ensuring that the moisture did not interfere with the spectra. If the moisture in the samples was not completely removed, interference appeared at $3500\,\mathrm{cm^{-1}}$, $2100\,\mathrm{cm^{-1}}$, and $1640\,\mathrm{cm^{-1}}$. A measurement time of approximately $5\,\mathrm{min}$ ($\sim 300\,\mathrm{s}$) did not cause any degradation of the components.

Each ATR-FTIR spectral dataset of a blood sample contained 1738 data points in the spectral range $4000–650\,\mathrm{cm^{-1}}$ (Fig 1(B)). Three spectral ranges, $3000–2800\,\mathrm{cm^{-1}}$, $1800–1650\,\mathrm{cm^{-1}}$,

and 1500–1350 cm$^{-1}$, of characteristic lipid absorption were selected as inputs. The characteristic absorption bands of the blood samples for subsequent analysis included absorption bands (1) in the spectral range of 3000–2800 cm$^{-1}$ and 1500–1350 cm$^{-1}$ assigned to the methyl and methylene groups of lipid molecules, (2) at ~ 1740 cm$^{-1}$ for the carbonyl group of lipids, (3) at ~ 1650 cm$^{-1}$ assigned to C = O stretching of the peptide bond, and (4) in the spectral range of 1500–1000 cm$^{-1}$ for the phosphodiester groups of nucleic acids, phospholipids, and side chains of peptides, and glycans [20–23] as shown in Fig 1(B).

## ATR-FTIR spectral analysis

All analyses and model calculations were performed using Python 3.9, NumPy, scikit-learn, imbalanced-learn, and matplotlib packages. The ATR-FTIR spectral data were used mainly to (1) determine the presence of MetS (classification task) and (2) predict the blood levels of several lipids (regression task) as follows: TG, total cholesterol (CHOL), HDL-C, low-density lipoprotein-cholesterol (LDL-C), and very-low-density lipoprotein cholesterol (VLDL-C). In conventional data analysis, data points that deviate significantly from the distribution are typically identified as outliers and eliminated to optimize the performance of the models. However, in this study, data outliers were not removed because all blood lipid levels were obtained from a trustworthy and standardized laboratory at the medical center.

## Models used for data preparation

Based on the previously reported prevalence of MetS, approximately 33% [23] oversampling was performed to generate a balanced dataset. An adaptive synthetic approach (ADASYN) [24] was used to generate calibration data. The performance and credibility of the model were assessed using 10-fold cross-validation.

## Models used for data analysis

The models used in the regression task include basic linear regression (LR), gradient boosting regression tree (GBTR), random forest regression tree, and histogram-based gradient boosting tree classifier (HGBTR), which have the advantage of starting the estimation quickly without loss of accuracy [25]. For the classification task, that is, the determination of MetS, logistic regression, random forest tree classifier (RF), gradient boosting tree classifier (GBT), and histogram-based gradient boosting tree classifier (HGBT) [25]. The model for each task was individually tested to determine the best target prediction results.

## Metrics for model explanation

The relationships between the actual laboratory values and model-predicted values for each prediction target were presented in the residual plot to evaluate the regression task results, along with the calculated values of R squared ($R^2$), mean absolute error (MAE), and mean absolute percentage error (MAPE), which were calculated by taking the average of the absolute differences between the predicted and actual values, divided by the actual values, and then expressed as a percentage. This can provide a clear and interpretable measure of prediction accuracy by expressing errors as a percentage of actual values. Subsequently, 10-fold cross-validation was used to calculate the mean and standard deviation (SD) of the above metrics across 10 models. A smaller standard deviation suggests that the performance of the model is relatively consistent across different folds.

The receiver operating characteristic curve (ROC) and confidence intervals (CI) were presented for 10-fold cross-validation to evaluate the classification task results. Other evaluation

metrics included area under the curve (AUC), F1, precision, and recall. The mean and SD were also calculated.

The results of the regression models were plotted as the weights of the best-performing GBT model over a 10-fold range. The scaled weighting was from −1 to 1, greater than 0.5, or smaller than −0.5 for the selected absorption wavenumbers for further evaluation.

To determine MetS, t-distributed stochastic neighbor embedding (TSNE) [26] was used to verify the data distribution in a high-dimensional 2D plane. Regarding the model explanation for the classification task, we applied the SHAP (SHapley Additive exPlanations) values with a tree-based explainer [27] to determine the nine wavenumbers with the greatest impact. This method enabled us to assess the consistency between our findings and those in the current literature.

## Results

### Study subjects

This study included 25 non-MetS and 25 MetS patients (Table 1). The two groups were age- and sex-matched. The MetS group had higher incidences of hypertension, diabetes mellitus, body weight, waist circumference, and hip circumference. In the MetS group, 56% of the subjects were taking lipid-lowering drugs. The mean MetS score was 3.96 for the MetS group and 0.32 for the non-MetS group. In the laboratory data, the MetS group had greater TG, VLDL-C, non-HDL-C, uric acid, alanine aminotransferase (AST), and aspartate aminotrans-ferase (ALT). HDL-C was significantly lower in the MetS group (Table 1). Upon examining the distribution of blood lipid values, we discovered that some patients with MetS exhibited extreme TG, VLDL, and LDL values. Prior to running the model, our data analysis indicated significant differences between the MetS and non-MetS groups for all parameters except CHOL (more information in S1 Appendix).

### ATR-FTIR spectral data characteristics

Spectral data were collected following a standard calibration procedure, using 150 blood sam-ples from 50 individuals. Each blood sample was repeatedly used to collect FTIR spectral data to generate 491 FTIR data (300 from MetS and 191 from non-MetS subjects). The ADASYN method was used to match the prevalence balance with 33% of MetS cases, and the dataset numbers were 300 for MetS and 609 for non-MetS cases. We categorized lipid values into five equally divided groups from maximum to minimum to examine potential spectral differ-ences among varying lipid levels (S2 Appendix). The mean ATR-FTIR spectrum for each group is plotted as a line, with the shaded regions representing plus one standard deviation. This visualization highlights the trends in spectral differences across lipid groups. However, although these trends suggest potential relationships, they do not allow for direct quantitative conclusions or inferences. Overlapping or minimal differences in FTIR spectral absorption across lipid ranges further suggest that nonlinear associations may exist which are not directly observable in raw spectral data. All the figures are listed in S2 Appendix. Grouped Spectrum with Specified Lipid Value Ranges.

### Regression task results

The correlation between the predicted values derived from the different regression models and actual laboratory values is shown in Fig 2 and Table 2. Prediction errors (residuals = actual values – predicted values) were also examined for TG, CHOL, HDL-C, LDL-C, and VLDL-C (Fig 2A–2E). The best model for predicting all targets based on $R^2$ was HGBT,

**Table 1. Demographic and laboratory data.**

| Items | Non-MetS (n = 25) | MetS (n = 25) | P value |
|---|---|---|---|
| Female (n)/ Male (n) | 19/ 6 | 12/ 13 | 0.08 |
| Age (years), mean (SD) | 49.12 (6.66) | 52.84 (7.20) | 0.064 |
| MetS score, mean (SD) | 0.32 (0.75) | 3.96 (0.89) | <0.001 |
| Hypertension, n (%) | 0 (0.0) | 24 (96.0) | <0.001 |
| Diabetes mellitus, n (%) | 2 (8.0) | 15 (60.0) | <0.001 |
| Height (cm), mean (SD) | 161.66 (8.16) | 162.52 (7.46) | 0.699 |
| Body weight (Kg), mean (SD) | 58.68 (11.80) | 75.69 (12.67) | <0.001 |
| BMI (Kg/m$^2$), mean (SD) | 22.38 (3.59) | 28.52 (3.34) | <0.001 |
| Waist (cm), mean (SD) | 75.07 (9.23) | 94.60 (9.07) | <0.001 |
| Hip circumference (cm), mean (SD) | 94.58 (7.69) | 101.56 (7.10) | 0.002 |
| Systolic BP (mmHg), mean (SD) | 115.60 (9.34) | 147.88 (22.46) | <0.001 |
| Diastolic BP (mmHg), mean (SD) | 70.32 (7.51) | 90.20 (16.21) | <0.001 |
| Pulse rate (/min), mean (SD) | 73.56 (9.96) | 80.40 (13.26) | 0.045 |
| Use of lipid-lowering drugs[*] (%) | 0 (0.0) | 14 (56.0) | <0.001 |
| **Laboratory data** | | | |
| Cholesterol (mg/dL), mean (SD) | 194.28 (33.56) | 185.29 (30.90) | 0.329 |
| Triglyceride (mg/dL), mean (SD) | 89.82 (25.17) | 226.30 (133.96) | <0.001 |
| HDL-C (mg/dL), mean (SD) | 68.20 (12.03) | 41.16 (10.00) | <0.001 |
| LDL-C (mg/dL), mean (SD) | 122.59 (32.91) | 119.33 (28.02) | 0.708 |
| VLDL-C (mg/dL), mean (SD) | 5.90 (29.39) | 24.80 (19.16) | 0.01 |
| Non-HDL-C (mg/dL), mean (SD) | 69.62 (26.08) | 144.13 (28.25) | <0.001 |
| Glucose (mg/dL), mean (SD) | 85.41 (11.83) | 118.59 (31.18) | <0.001 |
| HbA1C (%), mean (SD) | 5.44 (0.32) | 6.41 (1.15) | 0.074 |
| BUN (mg/dL), mean (SD) | 14.02 (4.76) | 15.55 (4.11) | 0.231 |
| Creatinine (mg/dL), mean (SD) | 0.72 (0.14) | 0.84 (0.27) | 0.048 |
| AST (IU/L), mean (SD) | 20.20 (5.92) | 25.41 (10.12) | 0.031 |
| ALT (IU/L), mean (SD) | 17.48 (7.01) | 29.38 (16.52) | 0.002 |
| Uric acid (mg/dL), mean (SD) | 4.69 (0.99) | 6.19 (1.47) | <0.001 |

MetS, metabolic syndrome; BMI, body mass index; BP, blood pressure; HDL-C, high-density lipoprotein cholesterol; LDL-C, low-density lipoprotein cholesterol; VLDL-C, very-low-density lipoprotein cholesterol; HbA1c, hemoglobin A1c; BUN, blood urea; AST, aspartate aminotransferase; ALT, alanine aminotransferase.

[*]The lipid-lowering drugs include statins (atorvastatin, rosuvastatin, lovastatin), niacin, fenofibrate, and ezetimibe.

whereas the model with the best MAE value was RF (Table 2). In our analysis, we observed that deviance decreased across most of the folds during the cross-validation training and testing phases. However, a few folds exhibited irregular patterns, possibly due to outliers in the data distribution (S3 Appendix for more information).

In the residual evaluation (right panels of Fig 2A–2E), the performance of LR (shown by green dots) was poor ($R^2 < 0.3$) with large MAE values, whereas the performance of RF/GBT/HGBT was similar. Among the different lipid parameters, the prediction performance was the best for TG and the worst for LDL-C, particularly with the LR model. The widely used Friedewald formula for calculating LDL-C is known to have inherent errors when compared with direct blood measurements, although it has been reported to be the most accurate way to calculate fasting LDL-C in Chinese people [28]. For example, the MAPE and MAE for LDL-C, calculated using the Friedewald formula, were 77.5% and 92.3 mg/dL, respectively. In this

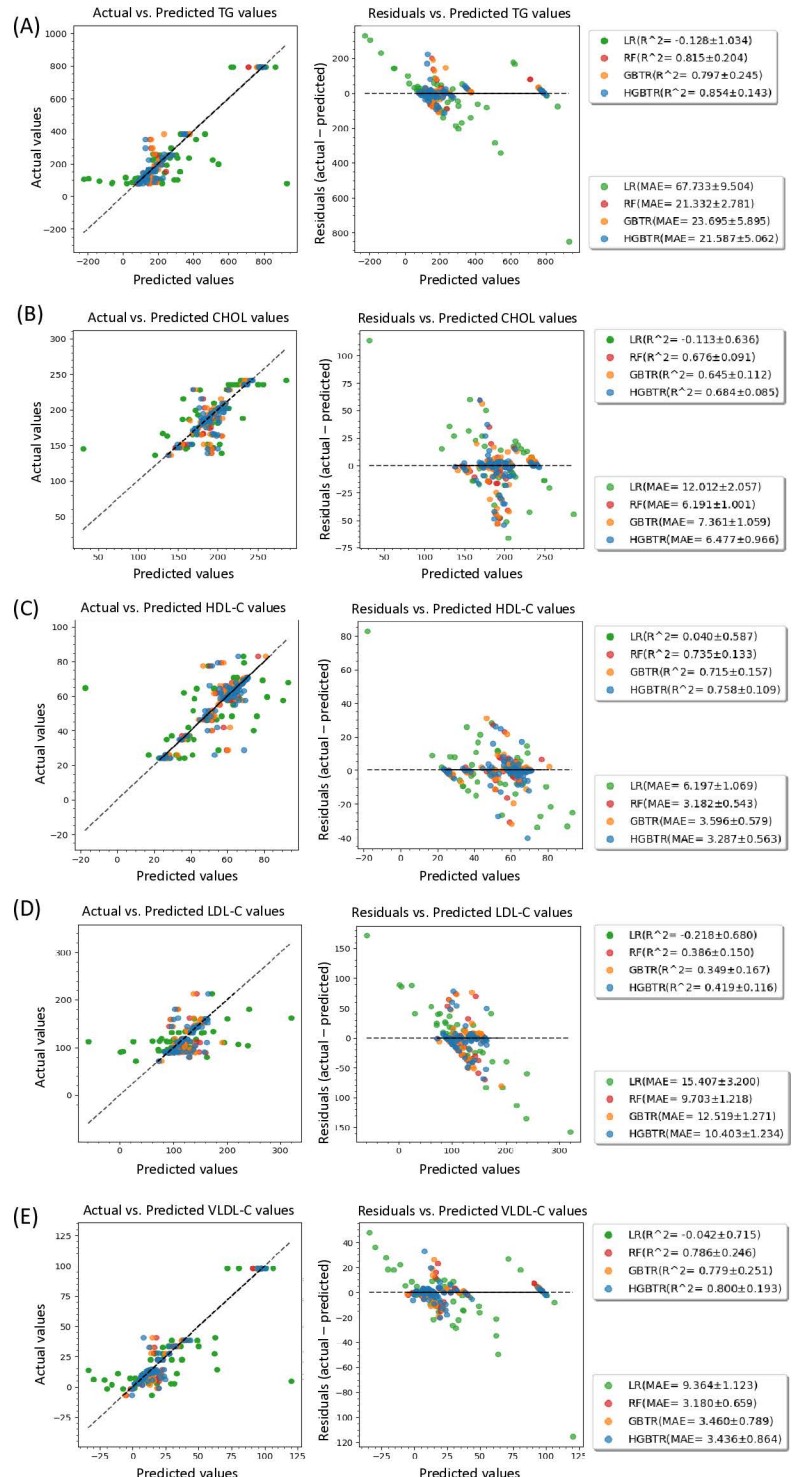

**Fig 2. FTIR spectrometry values predict the plasma concentrations of cholesterol (CHOL) and triglycerides (TG).** Correlation between the predictive values of FTIR spectrometry and the reference data (left panel). Residual plots (right panels) of IR absorbance versus the blood concentrations of (A)TG, (B) total cholesterol, (C) HDL-C, (D) LDL-C, and (E) VLDL-C.

**Table 2. The result of the regression task and classification task.**

**Regression task**

| | Models | LR | | | RF | | | GBT | | | HGBT | | |
|---|---|---|---|---|---|---|---|---|---|---|---|---|---|
| | Metrics | $R^2$ | MAE | MAPE (%) | $R^2$ | MAE | MAPE (%) | $R^2$ | MAE | MAPE (%) | $R^2$ | MAE | MAPE (%) |
| **Prediction targets** | **TG** | 0.13 (1.03) | 67.73 (9.50) | 41.9 | 0.82 (0.20) | **21.33** (2.78) | 12.3 | 0.80 (0.24) | 23.69 (5.89) | 14.1 | **0.85** (0.12) | 21.58 (5.06) | **12.5** |
| | **CHOL** | 0.11 (0.63) | 12.01 (2.05) | 6.9 | 0.68 (0.09) | **6.19** (1.00) | **3.4** | 0.65 (0.11) | 7.36 (1.05) | 3.7 | **0.68** (0.08) | 6.47 (0.96) | 3.5 |
| | **HDL-C** | 0.04 (0.58) | 6.19 (1.06) | 13.4 | 0.74 (0.13) | **3.18** (0.54) | **6.9** | 0.72 (0.15) | 3.59 (0.57) | 8.1 | **0.76** (0.10) | 3.28 (0.56) | 7.0 |
| | **LDL-C** | 0.22 (0.68) | 15.40 (3.20) | 12.2 | 0.39 (0.15) | **9.70** (1.21) | **7.3** | 0.35 (0.16) | 12.51 (1.27) | 10.0 | **0.42** (0.11) | 10.40 (1.23) | 7.9 |
| | **VDLD-C** | 0.04 (0.71) | 9.36 (1.12) | 109.9 | 0.79 (0.24) | **3.18** (0.65) | **38.0** | 0.78 (0.25) | 3.46 (0.78) | 42.1 | **0.80** (0.19) | 3.43 (0.86) | 41.6 |

**Classification task**

| | Precision | Recall | AUC | F1 | Accuracy |
|---|---|---|---|---|---|
| **LR** | 0.613 (0.11) | 0.809 (0.048) | 0.852 (0.047) | 0.694 (0.084) | 0.772 (0.046) |
| **RF** | 0.955 (0.037) | 0.833 (0.054) | 0.982 (0.01) | 0.889 (0.035) | 0.934 (0.014) |
| **GBT** | 0.938 (0.066) | 0.836 (0.038) | 0.971 (0.018) | 0.883 (0.033) | 0.930 (0.013) |
| **HGBT** | 0.931 (0.049) | 0.854 (0.059) | 0.976 (0.017) | 0.889 (0.032) | 0.932 (0.017) |

LR, linear regression; RF, random forest tree classifier; GBT, gradient boosting tree classifier; HGBT, histogram-based gradient boosting tree classifier; TG, triglyceride; HDL-C, high-density lipoprotein cholesterol; LDL-C, low-density lipoprotein cholesterol; VLDL-C, very-low-density lipoprotein cholesterol; CHOL, cholesterol; $R^2$, R squared; MAE, mean absolute error. AUC, area under the curve; F1, F1 score. All values are shown as the mean (SD). The best performances are indicated in bold.

study, our best model (Random Forest) achieves a significantly lower MAPE of 7.3% and an MAE of 9.7 mg/dL for LDL-C prediction.

Next, coefficient modeling was performed. The wavenumbers that were strongly correlated in predicting specific lipid parameters in the GBT model are shown in the heatmap (Fig 3). To predict TG, the data points were 2991, 1724, and 1722 cm⁻¹. For LDL-C, the data included points at 2933, 2931, 2906, and 1766 cm⁻¹. For HDL-C, the data included points at 2906, 2904, 2902, and 2898 cm⁻¹. For Chol were at 2931, 2916, 2906, and 1755 cm⁻¹. Furthermore, the data points were 2991 and 1722 cm⁻¹ for VLDL-C, respectively.

We further investigated the effect of MetS on the lipid value predictions of the model. The error distribution for our predictions showed that MetS subjects had more prediction errors than non-MetS subjects, indicating that the prediction accuracy was more accurate in the absence of MetS. To explore this further, we tested models trained and validated separately for MetS and non-MetS groups. The results indicated that the models trained on non-MetS data performed consistently well on non-MetS test data, whereas those trained on MetS data exhibited significant variability (**S3 Appendix** for more information).

These findings suggest that, while our model is adequate for general rapid scans, additional data and validation are necessary with a sufficient population size for patients with MetS. This also highlights the substantial variability in lipid disorders among patients with MetS. Moreover, FTIR provides an efficient and visualized screen to demonstrate lipid abnormalities in MetS (**S4 Appendix** for detailed information).

## Classification task results

These models were used to examine their ability to classify MetS and non-MetS groups, as shown in Fig 3 and Table 2. The best model was RF, with an AUC of 0.978, followed by HGBT (AUC, 0.972), GBT (AUC, 0.967), and LR (AUC, 0.851) (Fig 4A). The F1 score was similar,

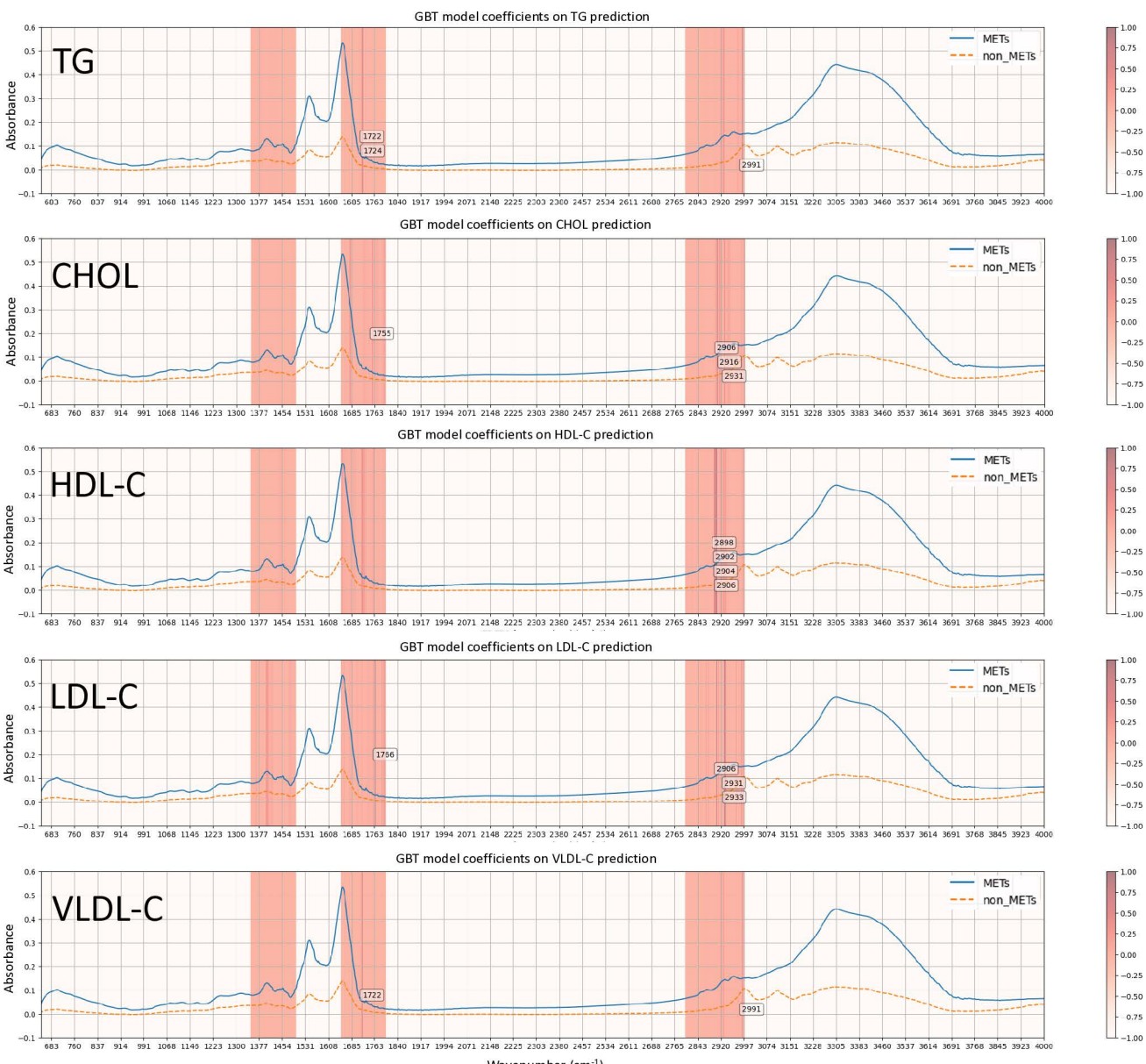

**Fig 3. Feature weights of the GBT prediction model for TG, CHOL, HDL-C, LDL-C, and VLDL-C.** The blue line represents the mean value of the FTIR spectra of total MetS, the mean value of the FTIR spectra of total MetS, and the orange dotted line represents the mean value of the FTIR spectra of total non-MetS. The degree of darkness in the vertical line indicates the magnitude of the correction. The unselected frequency bands are blank. The frequencies with an absolute coefficient larger than 0.5 are specifically labeled.

approximately around 0.88 for RF, HBGT, and GBT, but 0.69 for LR. The results suggest that RF, HBGT, and GBT can distinguish between MetS and non-MetS using ATR-FTIR data.

Projection verification of all the samples was performed using the t-SNE method (Fig 4B). Each projection point indicates high-dimensional data on the 2D plane; the color marks MetS, and the point density reflects the similarity of the data. In the 2D projection, the distinction between most MetS and non-MetS samples was well illustrated; however, a few data points remained relatively similar and overlapped. The excellent AUC and F1 ratings of all three

models (Table 2) indicate that MetS and non-MetS could be correctly classified using an appropriately constructed nonlinear model with FTIR spectral data. Concurrently, as shown in Fig 4C, the SHAP value analysis revealed that the top nine wavenumbers were 2843, 1496, 1653, 1699, 1655, 2891, 2935, 1410, and 2902 cm$^{-1}$. Notably, four wavenumbers were within the 2800–3000 cm$^{-1}$ range, two wavenumbers were within the 1350–1500 cm$^{-1}$ range, and three were within the 1600–1700 cm$^{-1}$ range. This finding indicates that when the task requires a comprehensive assessment rather than a single numerical value, the model considers all relevant lipid-associated frequency bands. SHAP waterfall plots identified bands that significantly contributed to MetS predictions (Fig 4D). For instance, wavenumbers of 1660, 1655, and 1657 cm$^{-1}$ were found to be the most significant contributors to the model's prediction of MetS (Fig 4E–G). These plots revealed that wavenumbers of 1653, 1655, and 1660 cm$^{-1}$ exhibited a stronger association with MetS predictions once they exceeded a certain threshold, further supporting the idea that FTIR detects long-chain lipid structures in the blood, which are crucial features for our model's predictions.

## Discussion

This study has proven our hypothesis that ATR-FTIR spectral data and appropriate machine learning methods can be used to accurately predict lipid levels in blood samples and distinguish between the absence and presence of MetS.

With the emergence of new lipid-lowering drugs based on various pharmaceutical targets, stratification of patients with MetS has become increasingly important. The more comprehensive the lipid disorders are determined, the more precisely the treatment will be chosen to optimize the efficacy and outcome benefits. Lipids are a broad group of organic molecules with structures that produce different infrared absorptions or emissions, which can be detected using FTIR spectrometry. FTIR spectrometry has been used to quantify the oil and fat content of food [29–31]. For instance, it precisely detects different lipid fractions and free fatty acids in oleaginous yeasts [30]. Bel'skaya et al. published their work on using model solutions in applying FTIR spectroscopy for the quantitative analysis of blood serum [32]. Portaccio et al. published their study results on ATR-FTIR spectra of commercial lipid samples and demonstrated specific fingerprint regions in ATR-FTIR spectra for phospholipids (phosphatidylcholine, phosphatidylethanolamine, phosphatidylserine, and phosphatidylinositol) from 3500 to 2800 cm$^{-1}$ and 1800 to 600 cm$^{-1}$, sphingolipids (ceramide, sphingosine 1-phosphate, ceramide 1-phosphate, and sphingomyelin) from 3500 to 2800 cm$^{-1}$ and 1800 to 600 cm$^{-1}$, and CHOL from 3150–2800 cm$^{-1}$.[33] Their study proved the usefulness of FTIR spectra for analyzing complex lipids, and this application can ultimately contribute to lipid research and its implications for human health. Similarly, FTIR could precisely predict TG, CHOL, and different-density lipoproteins in this study. The best spectral region(s) for predicting total TG were 2927, 2925, 1718, and 1500 cm$^{-1}$, 1718 cm$^{-1}$ for CHOL, and 2937 cm$^{-1}$ for HDL-C. Those were 2895, 2806, 2800, 1745, 1743, and 1716 cm$^{-1}$ for LDL-C, and were 2931 and 1500 cm$^{-1}$ for VLDL-C. The regions are close to those in a previous study, which evaluated 31 healthy individuals and suggested spectral ranges for lipids as 1741, 1747, 2854, and 2929 cm$^{-1}$ [34]. Jessen et al. used FTIR spectrometry to analyze blood plasma without any sample preparation to achieve stable and accurate measurements for concentrations of clinically relevant chemical constituents, including TG and CHOL [35]. Similarly, Li et al. used FTIR spectrometry to determine glucose and CHOL in whole blood and a partial least squares (PLS) regression fusion modeling method was used [36].

The choice of machine learning method affects the accuracy of the prediction of blood lipids, and this phenomenon was clearly demonstrated in this study. Linear and logistic

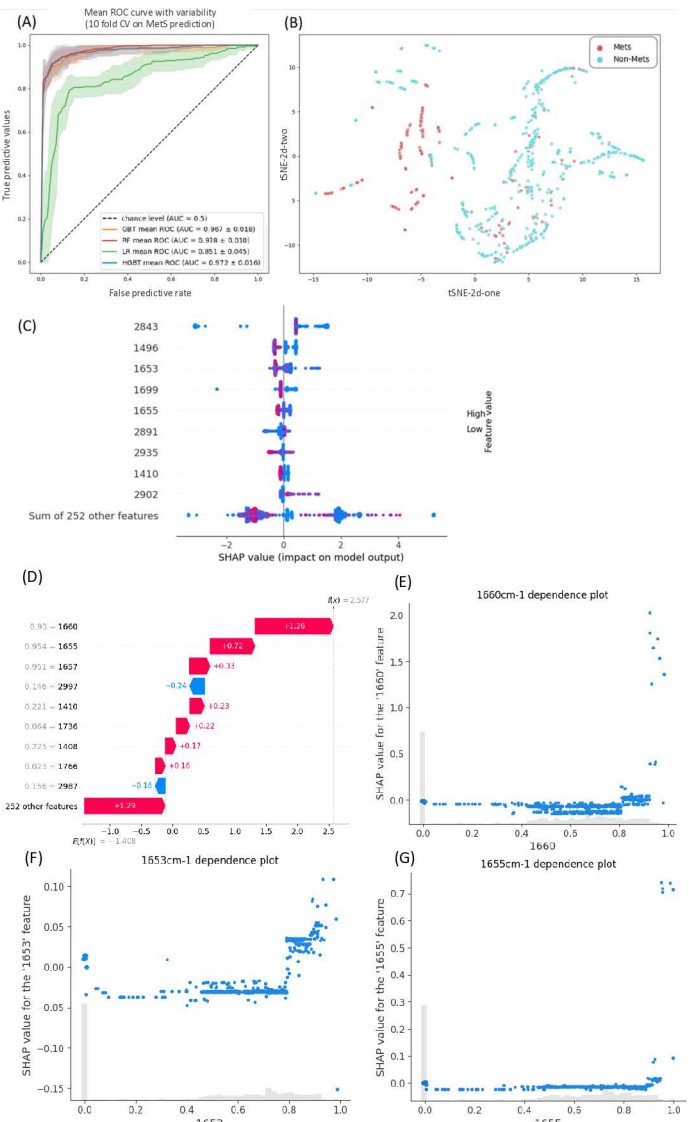

**Fig 4. MetS is distinct from non-MetS in the ATR-FTIR spectrometry analysis.** (A) ROCs curves of the different models in the MetS prediction task. The color bands represent the CI ranges. (B) The t-SNE method shows separated clustering between MetS and non-MetS spectrometry data. Light-blue dots represent non-MetS and red dots represent MetS. (C) The SHAP values for the GBT model in the MetS prediction task are presented in descending order, with features ranked from top to bottom based on their influence on prediction. (D) SHAP waterfall plots of the METS predictions. (E) SHAP partial-dependence plots for the absorption at 1660 cm⁻¹. (F) SHAP partial-dependence plots for the absorption at 1653 cm⁻¹. (G) SHAP partial-dependence plots for the absorption at 1655 cm⁻¹.

regression models perform poorly with high-dimensional data. It can be inferred that our model did not initially extract the features using PCA. However, RF, GPT, and HGPT continue to perform well for TG/CHOL/HDL-C, where a specific algorithm alone is sufficient to extract information from the original input and map the results.

The prediction performance also varied among different blood lipids. For LDL-C, which had a relatively wide concentration range, the $R^2$ value was relatively low. Nevertheless, the MAE value was between 9 and 12, and our model achieved a considerably low prediction error (as low as 7.3%) (Table 2). For instance, with an actual LDL level of 100 mg/dL, our model's

prediction ranged from 93.7 to 107.3 mg/dL. Previous studies suggested a desirable bias of ± 6.8% and a desirable total allowable error (TEa) of ± 13.7% for LDL laboratory data and calculated errors by analyzing the European Federation of Clinical Chemistry and Laboratory Medicine database [37]. Therefore, the prediction error for LDL-C in this study would not affect the medication decisions or clinical judgment. In the case of TG, all the machine learning models exhibited superior prediction accuracy, with $R^2$ values of 0.801–0.851 (Table 2).

Consistent with the existing knowledge, the model's interpretation revealed that the focus of HGBT lies between 3000 and 2800 cm$^{-1}$. We also attempted to restrict the input variables to the theoretical spectrum and entire spectral range in wavenumbers; however, the performance did not increase considerably, indicating that the interval chosen for this experiment was adequate. Using the high-dimensional projection technique, t-SNE, FTIR spectral analysis can effectively identify MetS samples. The SHAP values represent the contribution of each feature (in our case, each wavenumber) to the model's output relative to the average prediction (Fig 4). Essentially, SHAP values explain how a particular feature impacts the model's prediction, serving as a simplified, interpretable model to elucidate the workings of a more complex machine learning model. However, interpretability challenges can occasionally be introduced because the SHAP uses a linear approach to explain nonlinear models. Literature indicates that the range of 1645–1675 cm$^{-1}$ primarily corresponds to non-cyclic carbon-carbon double bonds (C = C), which may be related to the structure of long-chain lipids in the blood. Notably, the combined contribution of the other 252 parameters was greater than that of the single wavenumber at 1660 cm$^{-1}$, indicating the advantage of using SHAP in the FTIR spectrometry analysis.

This study has several limitations. Firstly, the limited sample size hindered our ability to evaluate the effectiveness of FTIR spectrometry in determining the severity of MetS. Secondly, although differences between plasma and serum FTIR spectra have been reported, this aspect was not investigated here. Lastly, the selection and exclusion of absorption bands in FTIR spectra may have introduced some degree of bias. Although the number of data items was sufficient for machine learning calculations, evaluating the model will require additional data. As we employed the ADASYN approach to balance the prevalence of the data, since additional data were created to compensate, the impact on actual applications must be further investigated. Fourth, verification using liquid chromatography-mass spectrometry (LC-MS) standards is lacking. The Taiwan Accreditation Foundation approved the laboratory for the fulfillment of ISO 15189 (Medical Laboratories Requirements for Quality and Competence), which supports the reliability of blood lipid concentrations. Future studies should comprehensively compare the accuracy, sensitivity, and specificity of the results from the AI-assisted FTIR spectrometry method against established methods with lipid standards. And we may explore causation-focused models to investigate molecular associations further and validate these predictions. However, given the study's current scope, our machine learning approach offers valuable insights into broad lipid level trends without requiring the extensive molecular characterization that lies beyond this study's objectives.

The developed AI-assisted FTIR spectrometry method is a nondestructive, reagent-free, and robust technology capable of accurately predicting lipid concentrations and identifying MetS in a prepared plasma sample. Macroscopic assessment is an unmet clinical requirement for diseases associated with metabolic disorders. This method provides a complete scout view of all MetS-related biochemical constituents, including all lipid species. Therefore, residual lipid-related risks can be revealed entirely. This comprehensive feature provides the potential for establishing an assessment system to stratify individuals with MetS to guide medical decisions and monitor therapeutic effects. In conclusion, AI-assisted FTIR spectrometry blood sample analysis is worth future large population studies to be applicable in clinical settings.

## Supporting information

**S1 Appendix. Distribution of blood lipid values in MetS and non-MetS groups.**
(PDF)

**S2 Appendix. Grouped spectrum with specified lipid value ranges.**
(PDF)

**S3 Appendix. Training deviance of the GBT model on different targets.**
(PDF)

**S4 Appendix. The effect of the presence of MetS on the model's lipid values prediction.**
(PDF)

**S5 File. The de-identified supporting data.**
(CSV)

## Acknowledgments

The authors thank Miss Wun-Jyun Jhuang for her skillful handling of the blood sample preparation and on the ATR-FTIR platform. The authors also thank Dr. Yi-Hsiung Lin for his assistance with research coordination at the beginning.

## Author contributions

**Conceptualization:** Tz-Ping Gau, Yao-Chang Lee, Hsiang-Chun Lee.

**Data curation:** Tz-Ping Gau, Jen-Hung Wen, Pei-Yu Huang, Hsiang-Chun Lee.

**Formal analysis:** Tz-Ping Gau, Jen-Hung Wen, I-Wei Lu, Wei-Po Lee.

**Funding acquisition:** Hsiang-Chun Lee.

**Investigation:** Pei-Yu Huang, Yao-Chang Lee, Hsiang-Chun Lee.

**Methodology:** Tz-Ping Gau, Pei-Yu Huang, Yao-Chang Lee, Wei-Po Lee, Hsiang-Chun Lee.

**Project administration:** Hsiang-Chun Lee.

**Resources:** Hsiang-Chun Lee.

**Software:** Tz-Ping Gau, I-Wei Lu, Wei-Po Lee.

**Supervision:** Hsiang-Chun Lee.

**Validation:** Tz-Ping Gau, Yao-Chang Lee, Hsiang-Chun Lee.

**Visualization:** Tz-Ping Gau, Yao-Chang Lee, Hsiang-Chun Lee.

**Writing – original draft:** Tz-Ping Gau, Jen-Hung Wen, Hsiang-Chun Lee.

**Writing – review & editing:** Yao-Chang Lee, Hsiang-Chun Lee.

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
