## [Decision Letter · Decision Letter 0]

8 May 2024

PONE-D-24-11405Application of Attenuated Total Reflection–Fourier Transform Infrared spectroscopy in semi-quantification of blood lipids and characterization of the metabolic syndromePLOS ONE

Dear Dr. Lee,

Thank you for submitting your manuscript to PLOS ONE. After careful consideration, we feel that it has merit but does not fully meet PLOS ONE’s publication criteria as it currently stands. Therefore, we invite you to submit a revised version of the manuscript that addresses the points raised during the review process.

Please improve the novelty statement and sampling handling explanation.

We look forward to receiving your revised manuscript.

Kind regards,

Muhammad Iqhrammullah, Ph.D

Academic Editor

PLOS ONE

4. In this instance it seems there may be acceptable restrictions in place that prevent the public sharing of your minimal data. However, in line with our goal of ensuring long-term data availability to all interested researchers, PLOS’ Data Policy states that authors cannot be the sole named individuals responsible for ensuring data access (http://journals.plos.org/plosone/s/data-availability#loc-acceptable-data-sharing-methods ).

Additional Editor Comments:

Please improve the novelty statement and sampling handling explanation.

Reviewers' comments:

Reviewer's Responses to Questions

**Comments to the Author**

1. Is the manuscript technically sound, and do the data support the conclusions?

Reviewer #1: Partly

Reviewer #2: Yes

2. Has the statistical analysis been performed appropriately and rigorously? 

Reviewer #1: No

Reviewer #2: Yes

3. Have the authors made all data underlying the findings in their manuscript fully available?

Reviewer #1: Yes

Reviewer #2: Yes

4. Is the manuscript presented in an intelligible fashion and written in standard English?

Reviewer #1: Yes

Reviewer #2: Yes

5. Review Comments to the Author

Reviewer #1: 1. After carefully reviewing the introduction section, I did not find a clear and explicit statement highlighting the novelty or newness of the study. The authors do not appear to directly state what makes their work novel compared to previous research in the field.

2. The authors emphasize the advantages of their ATR-FTIR method, such as rapid assay times and the need for small blood samples (0.5 μL). These benefits make the method potentially better for frequent monitoring and clinical applications compared to established techniques. Unfortunately, the authors lack a comprehensive comparison of the accuracy, sensitivity, and specificity of their approach against established methods. A deeper analysis of the performance metrics and potential limitations of their technique, in relation to existing gold standards, would better support their case for the usefulness and innovation of their method.

3. The conclusion should explicitly reiterate the potential impact of early detection and management of MetS on improving patient outcomes and quality of life, as highlighted in the introduction.

4. What do you mean by the brackets in table 2 as standard deviation (SD)?

5. However, there are a few minor points that could be improved in the reporting of the SHAP analysis: The authors could provide more details on how exactly the SHAP values were calculated (e.g., using the TreeExplainer for tree-based models like GBT) and what they represent (e.g., the contribution of each wavelength to the model's output relative to the average prediction). They could also discuss any potential limitations or caveats of using SHAP for model interpretation in this specific context.

6. From a purely statistical perspective, an R2 value of 0.419 indicates that only 41.9% of the variance in LDL-C levels can be explained by the ATR-FTIR spectral data using the HGBT model. This suggests that the model's predictions for LDL-C are not as accurate as those for other lipid parameters like TG (R2 = 0.854) or HDL-C (R2 = 0.758). In many scientific contexts, an R2 value below 0.5 would be considered poor model performance. I'm uncertain about the authors' argument, which claims that despite the low R² value, the mean absolute error (MAE) for LDL-C prediction was between 9 and 12 mg/dL.

Reviewer #2: Interesting work that discusses the application of ATR-FTIR in biological samples. The study is needed and the use of AI algorithms added value. Just some comments are needed to clarify:

-How were the samples stored?

-How did the authors ensure stability of samples?

-Was any heating of the sample applied prior to measurement? To prevent water interference?

-Comment on transferability of the models to other matrices in the discussion

6. PLOS authors have the option to publish the peer review history of their article (what does this mean? ). If published, this will include your full peer review and any attached files.

**Do you want your identity to be public for this peer review?** For information about this choice, including consent withdrawal, please see our Privacy Policy .

Reviewer #1: No

Reviewer #2: No

---

## [Author Response · Author response to Decision Letter 0]

28 Jul 2024

28-July-2024

Editor-in-Chief

PLOS ONE

Dear Editors:

We would like to thank all the reviewers and editors for their comments on the manuscript (PONE-D-24-11405) entitled “Application of attenuated total reflection–Fourier transform Infrared spectroscopy in semi-quantification of blood lipids and characterization of metabolic syndrome”.

Following the reviewers’ comments, we revised the manuscript and provided responses to describe the manuscript amendments. Point-to-point responses have been included in this submission.

We apologize that the grant information we provided in the ‘Funding Information’ and ‘Financial Disclosure’ sections do not match. One of our authors forgot to add a Financial Disclosure Statement before the primary submission. We received funding to support this study and updated the Funding Information on the Submission page as follows: National Health Research Institutes (NHRI-EX107-10724SC, NHRI-EX108-10724SC, NHRI-EX109-10724SC, and NHRI-EX110-10724SC), Kaohsiung Medical University Hospital (KMUH111-1R07 and KMUH110-0R11), Taiwan Ministry of Science and Technology/National Sciences, and Technology Council Grants (MOST 109-2314-B-037-111-MY3) NSTC 112-2314-B037-069. Please change the online submission form on our behalf. Thank you.

Regarding raw data sharing, all authors have decided on a data-sharing plan, and our entire data will be made freely accessible if our manuscript is accepted for publication. We will adhere to your open-data policy. To meet your requirement that all data presented in the study be made publicly available at or before the time of acceptance, the file named “encoded_data” is uploaded to the Submission System as a Supporting Information.

We believe that, after incorporating the valuable reviewers’ and editors’ opinions, the manuscript is worthy of consideration for publication in PLOS ONE.

Sincerely yours,

Hsiang-Chun Lee, MD, PhD

Professor, College of Medicine, Kaohsiung Medical University

Attending Physician, Division of Cardiology, Department of Internal Medicine, Kaohsiung Medical University Hospital

100 Tzyou 1st Rd, Kaohsiung 807, Taiwan

Tel.: +886-7-3121101#7741 Fax: +886-7-3234845

E-mail: hclee@kmu.edu.tw

---

## [Decision Letter · Decision Letter 1]

5 Nov 2024

PONE-D-24-11405R1Application of attenuated total reflection–Fourier transform Infrared spectroscopy in semi-quantification of blood lipids and characterization of metabolic syndromePLOS ONE

Dear Dr. Lee,

Thank you for submitting your manuscript to PLOS ONE. After careful consideration, we feel that it has merit but does not fully meet PLOS ONE’s publication criteria as it currently stands. Therefore, we invite you to submit a revised version of the manuscript that addresses the points raised during the review process.

We look forward to receiving your revised manuscript.

Kind regards,

Rajesh Kumar Singh, Ph.D.

Academic Editor

PLOS ONE

Journal Requirements:

**Comments to the Author**

1. If the authors have adequately addressed your comments raised in a previous round of review and you feel that this manuscript is now acceptable for publication, you may indicate that here to bypass the “Comments to the Author” section, enter your conflict of interest statement in the “Confidential to Editor” section, and submit your "Accept" recommendation.

Reviewer #3: All comments have been addressed

Reviewer #4: All comments have been addressed

2. Is the manuscript technically sound, and do the data support the conclusions?

Reviewer #3: Yes

Reviewer #4: (No Response)

3. Has the statistical analysis been performed appropriately and rigorously? 

Reviewer #3: Yes

Reviewer #4: Yes

4. Have the authors made all data underlying the findings in their manuscript fully available?

Reviewer #3: Yes

Reviewer #4: Yes

5. Is the manuscript presented in an intelligible fashion and written in standard English?

Reviewer #3: Yes

Reviewer #4: Yes

6. Review Comments to the Author

Reviewer #3: The study based on Application of attenuated total reflection–Fourier transform Infrared spectroscopy in

semi-quantification of blood lipids and characterization of metabolic syndrome is highly significant. However I would suggest to authors please expand this study by using molecular methods.

Reviewer #4: • Provide additional context on ATR-FTIR: Add a brief explanation about how ATR-FTIR spectroscopy works, specifically for readers less familiar with the technique. This could improve the accessibility of the study’s technical approach.

• Add a visual comparison: Consider adding a figure that visually compares the ATR-FTIR spectrum with traditional spectra of individual lipid classes or a figure showing predicted vs. actual lipid levels for better visual interpretation.

7. PLOS authors have the option to publish the peer review history of their article (what does this mean? ). If published, this will include your full peer review and any attached files.

**Do you want your identity to be public for this peer review?** For information about this choice, including consent withdrawal, please see our Privacy Policy .

Reviewer #3: No

Reviewer #4: No

---

## [Author Response · Author response to Decision Letter 1]

10 Dec 2024

PONE-D-24-11405R1

Application of attenuated total reflection–Fourier transform Infrared spectroscopy in semi-quantification of blood lipids and characterization of metabolic syndrome

PLOS ONE

Comments to the Author

Reviewer #3: The study based on Application of attenuated total reflection–Fourier transform Infrared spectroscopy in semi-quantification of blood lipids and characterization of metabolic syndrome is highly significant. However I would suggest to authors please expand this study by using molecular methods.

Response:

Thank you for your comments. We agree that molecular methods are also very important for applying non-destructive ATR-FTIR spectrometry in serum analysis, particularly for lipids. The following studies demonstrate the use of ATR-FTIR spectrometry to analyze blood samples. Bel’skaya et al. published their work on using model solutions in applying FTIR spectroscopy for the quantitative analysis of blood serum.[32] Portaccio et al. published their study results on ATR-FTIR spectra of commercial lipid samples and demonstrated specific fingerprint regions in ATR-FTIR spectra for phospholipids (phosphatidylcholine, phosphatidylethanolamine, phosphatidylserine, and phosphatidylinositol) from 3500 to 2800 cm−1 and 1800 to 600 cm−1), sphingolipids (ceramide, sphingosine 1-phosphate, ceramide 1-phosphate, and sphingomyelin) from 3500 to 2800 cm−1 and 1800 to 600 cm−1), and cholesterol (3150–2800 cm−1) [33]. Their study proved the usefulness of FTIR spectra for analyzing complex lipids, and this application can ultimately contribute to lipid research and its implications for human health. (Line 319-328)

Based on the known advantage of FTIR spectra in quantitative analysis and sensitivity to differentiate various lipids, our study approach was directed toward linkage to clinical application. Consistent with the aforementioned literature, our results revealed the reliability of ATR-FTIR for semi-quantitative lipid analysis of human serum samples.

Unlike those publications, our study emphasizes the predictive capability of machine learning models. The core concept is to uncover the correlations between FTIR spectra, clinically derived lipid values, and diagnostic classifications.

Since this approach centers on data-driven correlations rather than direct molecular causation, it inherently lacks molecular specificity, which remains a limitation of ML-based predictions.

In future work, we may explore causation-focused models to investigate molecular associations further and validate these predictions. However, given the study’s current scope, our ML approach offers valuable insights into broad lipid level trends without requiring the extensive molecular characterization that lies beyond this study’s objectives. (Line 380-384)

Two new citations have been added.

32. Bel’skaya LV, Sarf EA, Solomatin DV. Application of FTIR Spectroscopy for Quantitative Analysis of Blood Serum: A Preliminary Study. Diagnostics. 2021;11(12):2391. PubMed PMID: doi:10.3390/diagnostics11122391.

33. Portaccio M, Faramarzi B, Lepore M. Probing Biochemical Differences in Lipid Components of Human Cells by Means of ATR-FTIR Spectroscopy. Biophysica. 2023;3(3):524-38. PubMed PMID: doi:10.3390/biophysica3030035.

 Reviewer #4: • Provide additional context on ATR-FTIR: Add a brief explanation about how ATR-FTIR spectroscopy works, specifically for readers less familiar with the technique. This could improve the accessibility of the study’s technical approach.

Response:

Thank you for your comments. We have added a paragraph to the Introduction and Methods section to briefly explain how ATR-FTIR spectroscopy works. The context is as follows.

ATR-FTIR spectroscopy is a technique for analyzing the chemical composition of materials by detecting their characteristic absorption of mid-infrared light. In ATR-FTIR, molecules absorb mid-infrared light at specific wavenumbers (or frequencies), depending on their chemical bonds and structure. Each type of bond (e.g., C=O and C–H) has a unique absorption pattern, creating a molecular fingerprint. (Line 77-81)

This study applied a small sample volume (0.5 µL to a Ge ATR crystal, as shown in Figure 1(A). A modulated mid-infrared beam was directed into the Ge crystal, generating an evanescent wave at the crystal surface, allowing the light to interact with the sample. The sample absorbs light at characteristic wavenumbers, and the remaining light is detected using a liquid nitrogen-cooled mercury cadmium telluride (MCT) infrared detector. The resulting spectrum shows peaks corresponding to specific molecular bonds, providing a unique chemical profile of the sample. (Line 122-127)

Figure 1(A). The schematic depicts the attenuated total reflection for acquiring the FTIR spectra of a blood sample. The mid-infrared light is modulated using the Michaelson interferometer of an FTIR spectrometer. (Line 142-144)

• Add a visual comparison: Consider adding a figure that visually compares the ATR-FTIR spectrum with traditional spectra of individual lipid classes or a figure showing predicted vs. actual lipid levels for better visual interpretation.

Response:

Thank you for your comments. To illustrate the ability of spectral values to quantify traditionally categorized lipid classes (e.g., cholesterol and triglycerides), we categorized lipid values into five equally divided groups from maximum to minimum to examine potential spectral differences among varying lipid levels (see Figures below). The X-axis represents the input ATR-FTIR wavenumbers, the Y-axis represents the input ATR-FTIR spectral values, and the color group corresponds to the measurement group obtained from conventional biochemical assays. The average ATR-FTIR spectrum for each group is shown as a line, with the shaded regions representing plus one standard deviation. This visualization highlights the trends in spectral differences across lipid groups. However, although these trends suggest potential relationships, they do not allow for direct quantitative conclusions or inferences. Overlapping or minimal differences in FTIR spectral absorption across lipid ranges further suggest that nonlinear associations may exist that are not directly observable in raw spectral data. By applying machine learning models, we can efficiently capture and interpret these complex patterns, offering precise and meaningful lipid-level predictions beyond what is visually discernible. All figures are listed in Appendix 2. Grouped Spectrum with Specified Lipid Value Ranges (Line 211-223)

In response to your suggestion, we have also included Figure 2 to illustrate the comparison of predicted versus actual lipid levels. In this figure, the x-axis represents the lipid values predicted by the model based on the input ATR-FTIR spectral data, whereas the y-axis shows the actual measured lipid values. Additionally, we included a visualization of the distribution of error values to further elucidate the prediction accuracy and deviations of the model.

---

## [Editor Report · Decision Letter 2]

13 Dec 2024

Application of attenuated total reflection–Fourier transform Infrared spectroscopy in semi-quantification of blood lipids and characterization of metabolic syndrome

PONE-D-24-11405R2

Dear Dr. Lee,

We’re pleased to inform you that your manuscript has been judged scientifically suitable for publication and will be formally accepted for publication once it meets all outstanding technical requirements.

Kind regards,

Rajesh Kumar Singh, Ph.D.

Academic Editor

PLOS ONE
---

## [Editor Report · Acceptance letter]

PONE-D-24-11405R2

PLOS ONE

Dear Dr. Lee,

I'm pleased to inform you that your manuscript has been deemed suitable for publication in PLOS ONE. Congratulations! Your manuscript is now being handed over to our production team.

Kind regards,

on behalf of

Dr. Rajesh Kumar Singh

Academic Editor

PLOS ONE